# *Cromileptes altivelis* microRNA Transcriptome Analysis upon Nervous Necrosis Virus (NNV) Infection and the Effect of cal-miR-155 on Cells Apoptosis and Virus Replication

**DOI:** 10.3390/v14102184

**Published:** 2022-10-03

**Authors:** Hehe Du, Zhenjie Cao, Zhiru Liu, Guotao Wang, Ying Wu, Xiangyu Du, Caoying Wei, Yun Sun, Yongcan Zhou

**Affiliations:** 1State Key Laboratory of Marine Resource Utilization in South China Sea, Hainan University, Haikou 570228, China; 2Collaborative Innovation Center of Marine Science and Technology, Hainan University, Haikou 570228, China; 3Hainan Provincial Key Laboratory for Tropical Hydrobiology and Biotechnology, College of Marine Science, Hainan University, Haikou 570228, China

**Keywords:** *Cromileptes altivelis*, miRNA, nervous necrosis virus (NNV), cal-miR-155, cell apoptosis

## Abstract

MicroRNAs (miRNAs) could regulate various biological processes. Nervous necrosis virus (NNV) is one of the primary germs of the Humpback grouper (*Cromileptes altivelis*), a commercial fish of great importance for Asian aquaculture. However, there is limited available information on the host-virus interactions of *C. altivelis*. miRNAs have been shown to play key roles in the host response to infection by a variety of pathogens. To better understand the regulatory mechanism of miRNAs, we constructed miRNA transcriptomes and identified immune-related miRNAs of *C. altivelis* spleen in response to NNV infection. Reads from the three libraries were mapped onto the *Danio rerio* reference genome. As a result, a total of 942 mature miRNAs were determined, with 266 known miRNAs and 676 novel miRNAs. Among them, thirty-two differentially expressed miRNAs (DEmiRs) were identified compared to the PBS control. These DEmiRs were targeted on 895 genes, respectively, by using miRanda v3.3a. Then, 14 DEmiRs were validated by qRT-PCR and showed consistency with those obtained from high-throughput sequencing. In order to study the relationship between viral infection and host miRNA, a cell line from *C. altivelis* brain (CAB) was used to examine the expressions of five known DEmiRs (miR-132-3p, miR-194a, miR-155, miR-203b-5p, and miR-146) during NNV infection. The results showed that one miRNA, cal-miRNA-155, displayed significantly increased expression in response to the virus infection. Subsequently, it was proved that overexpression of cal-miR-155 enhanced cell apoptosis with or without NNV infection and inhibited virus replication in CAB cells. Oppositely, the cal-miRNA-155 inhibitor markedly suppressed apoptosis in CAB cells. The results of the apoptosis-related genes mRNA expression also showed the regulation of cal-miR-155 on the apoptosis process in CAB cells. These findings verify that miR-155 might exert a function as a pro-apoptotic factor in reply to NNV stimulation in CAB cells and help us further study the molecular mechanisms of the pathogenesis of NNV in *C. altivelis*.

## 1. Introduction

Nervous necrosis virus (NNV) is the pathogen of viral nervous necrosis (VNN), which was first reported in Australia in the late 1980s and caused the death of many kinds of farmed marine fish species and serious economic loss worldwide since then [1,2]. It was then found that NNV belongs to the Nodaviridae family and is a non-enveloped, icosahedral RNA virus, which consists of two single-stranded positive-sense RNA molecules, RNA1 (3.1 kb) and RNA2 (1.4 kb) [3,4]. VNN is responsible for 100% mortality in most cultured groupers, and the fish seeding industry also loses as a result of VNN attacks in seed stadiums and even in adults. Humpback groupers (*Cromileptes altivelis*) are an important commodity due to their high economic value. Yet, among numerous kinds of diseases, VNN is one which causes a major loss to fish farmers [5,6]. The mystery of NNV has been gradually revealed, but more is still needed.

MicroRNAs (miRNAs), a kind of small non-coding RNA molecule, are usually known to have the function of down-regulating gene expression at the post-transcriptional level. A number of studies have found that miRNAs are crucially important in many biological processes, including but not limited to cell proliferation, cell apoptosis, signal transduction, and tumorigenesis [7,8]. Viral infection alters host miRNA expression, and in turn, host miRNA regulates viral infection [9]. Moreover, miRNAs are also closely related to immunity [10]. Whole miRNA transcriptome profiling analysis by rapid development high-throughput sequencing technologies provides an efficient method for understanding the genetic response of a host to diseases and pathogens. In recent years, researchers have conducted many high-throughput miRNA sequencing experiments of teleost fish or fish cells in response to various viral infections, such as Singapore grouper iridovirus (SGIV), megalocytivirus, viral hemorrhagic septicemia virus (VHSV), and NNV [11,12,13,14,15]. According to these sequencing results, many differentially expressed miRNAs (DEmiRs) were found. Among them, some have been proved to play critical roles in suppressing the inflammatory response and antiviral processes. In miiuy croaker, miR-128, miR-192, miR-145, and miR-375 have been proven to play critical roles in suppressing inflammatory responses by regulating the NF-κB signaling pathway via targeting different genes [16,17,18,19].

However, limited information on the genomics and suitable cell line of *C. altivelis* has hampered the understanding of the host-virus interactions molecular mechanisms. In this study, high-throughput sequencing was used to identify miRNAs involved in NNV infection progression. Among the DEmiRs we found, cal-miRNA-155 expression significantly increased in response to the virus infection. Furthermore, cal-miR-155 was found to have effects on the regulation of the apoptosis process and the inhibition of the replication of the virus in *C. altivelis* cells, indicating cal-miR-155 may function as a pro-apoptotic response to NNV stimulation. This research helps us better understand the role of miRNAs in the teleost fish immune system and offers new insight into the virus-cell interaction.

## 2. Materials and Methods

### 2.1. Virus and Cell Line

The NNV belonging to the red-spotted grouper nervous necrosis virus (RGNNV) genotype was derived from diseased juvenile *C. altivelis*, which was ensured to be infected by the NNV. The viral suspension was obtained from diseased fish tissue by homogenization in 10 volumes of PBS, centrifugated at 2000× *g* for 20 min at 4 °C. The supernatant was filtered through a 0.22 μm sterile filter to eliminate the interference of bacteria. The viral suspension was stored at −80 °C until further use. Inactivated NNV suspension was obtained by exposure under ultraviolet (UV) light for 4 h and shaking every 30 min on a clean bench as previously described with some modifications [20]. The UV-treated inactivation effect on the virus was verified by PCR. 

The CAB (*C. altivelis* brain) cell line was established by our lab and cultured in Leibovitz’s L-15 medium (Gibco, Grand Island, NY, USA) supplemented with 15% fetal bovine serum (FBS, Gibco) at 26 °C [21]. The viral stock was titrated in CAB cells by 50% tissue culture infectious dose (TCID_50_) determined as described previously [22]. In this study, CAB cells were infected with NNV with a multiplicity of infection (MOI) of 10 and incubated at 26 °C in the L-15 medium with 5% FBS.

### 2.2. Fish Acclimation, Toxicity Test, and Sample Collection

*C. altivelis*, with lengths of about 15 cm, were purchased from a fish breeding farm in Danzhou (Hainan Province, China). Before experimentation, fish were cultured for two weeks to adapt to the laboratory environment. The liver, head kidney, and spleen were randomly sampled from 5% of the fish for pathogen examination as reported before [23]. No bacteria or viruses were detected from the examined fish. *C. altivelis* were challenged by intraperitoneal injection with 0.1 mL of the NNV suspension (1 × 10^5^ TCID_50_/mL). Meanwhile, an equal volume of PBS was injected as a control. After three- and eight-days post-injection, representing the incubation period and the onset period of the virus in our pre-experiment, the spleen tissues of six fish from each group were sampled and pooled together as one sample for miRNA sequencing. Every group was replicated for three samples. Before tissue collection, fish were euthanized with tricaine methanesulfonate (Sigma, St. Louis, MO, USA).

In this study, all experiments involving live fish complied with the regulations for the Administration of Affairs Concerning Experimental Animals promulgated by the Animal Ethics Committee of Hainan University.

### 2.3. miRNA Library Construction and miRNAs Dentification

The total RNA was isolated from the spleens of the control and infected fish using TRIzol reagent (Life Technologies, Carlsbad, CA, USA) according to the manufacturer’s protocol, and miRNA construction was performed as previously described [13]. To obtain clean read libraries, the miRNA raw data was filtered as follows: removal of contaminated RNA with an adaptor, poly-A, low-quality reads, and length kept between 17 nt to 35 nt. Then, the rRNAs, tRNAs, snRNAs, and snoRNAs were removed. We aligned the clean reads in GenBank and Rfam11. 0, followed by using Bowtie (Version 1.1.1) (Hopkins, Baltimore, MD, USA) to compare each read to the reference genome (mismatch was set to less than or equal to 1), counted the number and percentage of reads, and filtering out the sequences that could not be compared with the reference genome [24,25]. The sequences of known miRNAs were obtained from the miRBase database (http://www.mirbase.org/) (Release 22.1, accessed on 29 October 2020), and the mirDeep2 software (Version 2.0.0.8) (Systems Biology group at the Max Delbrück Center, Berlin, Germany) was used to identify the known miRNAs [26]. miRNA which did not belong to any miRNA family were represented by “unknown” and called novel miRNAs. Novel miRNA secondary structure predictions were made, with hairpin structures predicted using RNAfold WebServer online software (http://rna.tbi.univie.ac.at//cgi-bin/RNAWebSuite/RNAfold.cgi) (accessed on 26 January 2022) based on the reference genome, and the optimal secondary structures with minimum free energy were calculated [27].

### 2.4. miRNAs Sequence Phylogenetic Evolution Analysis

For the purpose of studying the origin and evolution of the miRNAs we obtained from *C. altivelis*, miRNAs data, including their family classification from 25 species, were downloaded from miRBase. Their time tree was built depending on the evolutionary timescale of life by TIMETREE 5 (http://timetree.org/) (accessed on 13 June 2022) [28]. These species included two cephalochordata: *Branchiostoma belcheri* (*bbe*) and *Branchiostoma floridae* (*bfl*); eight fish: *Salmo salar* (*ssa*), *Paralichthys olivaceus* (*pol*), *Oryzias latipes* (*ola*), *Fugu rubripes* (*fru*), *Tetraodon nigroviridis* (*tni*), *Ictalurus punctatus* (*ipu*), *Danio rerio* (*dre*), and *Cyprinus carpio* (*ccr*); two amphibians: *Xenopus laevis* (*xla*) and *Xenopus tropicalis* (*xtr*); four mammals: *Monodelphis domestica* (*mdo*), *Mus musculus* (*mus*), *Homo sapiens* (*hsa*), and *Ornithorhynchus anatinus* (*oan*); two reptiles: *Anolis carolinensis* (*aca*) and *Ophiophagus hannah* (*oha*); two aveses: *Gallus gallus* (*gga*) and *Taeniopygia guttata* (*tgu*); two uro-chordatas (uch): *Oikopleura dioica* (*odi*) and *Ciona savignyi* (*csa*); two non-chordatas (nch): *Caenorhabditis elegans* (*cel*) and *Drosophila melanogaster* (*dme*); one cyclostomata: *Petromyzon marinus* (*pma*).

### 2.5. Differential Expression miRNAs Analysis and Prediction, Annotation, and Enrichment of Target Genes

After estimating the expression profiles of miRNA by TPM (transcript per million), the differential expression miRNAs analyses and P value adjustments were performed using the DESeq and Benjamini Hochberg method, respectively [29,30,31]. DEmiRs were screened with *p* value ≤ 0.05, |log2 (fold change) | ≥1 as the threshold. MiRanda (version 3.3a) (Memorial Sloan-Kettering Cancer Center, New York, NY, USA) was used to predict miRNA target genes [32,33]. After selecting target genes, their GO (Gene Ontology) enrichment and KEGG (Kyoto Encyclopedia of Genes and Genomes) pathway analysis were performed. Their relationship is not a simple one-to-one relationship, but a complex many-to-many relationship. The interaction network between miRNA and gene was constructed through association analysis, and the key miRNA and target genes were identified by network analysis.

### 2.6. Validation of DEmiRs by qRT-PCR

We selected 14 DEmiRs, including 5 known and 9 novel DEmiRs, and analyzed their relative expression by quantitative real-time PCR (qRT-PCR). The cDNA generation using 1 μg of total RNA, which was leftover post miRNA library construction, was performed using the miRNA 1st Strand cDNA Synthesis kit (by stem-loop) (Vazyme, Nanjing, China) following the instruction from the manufacturer. Briefly, the total RNA was first added to gDNA Wiper Mix to remove the gDNA at 42 °C for 2 min, then stem-loop primers (2 µM), 10 × RT Mix, and HiScript II Enzyme Mix were added to the tubes. The stem-loop primers are shown in Appendix A, and designed by the miRNA Design software from Vazyme (Nanjing, China). U6 was selected as the reference gene control [34].

### 2.7. Virus Infection and the Expressions of the DEmiRs in CAB Cells

The NNV infection in the CAB cell line was performed as follows. First, the cells were digested and diluted to a cell concentration at 6 × 10^4^ cell/mL, then 2 mL of this suspension was added into a 6-well cell culture plate and incubated for 24 h at 26 °C to approximately 70% confluence. UV-inactivated viral suspension was added simultaneously as a control. After NNV suspension was adsorbed for 2 h (MOI = 10), the medium containing NNV was discarded, and the cells were maintained in low serum (5% FBS) L-15 media at 26 °C. At the time-points 24, 48, and 72 h post-infection (hpi), virus-specific cytopathic effects (CPE) were observed by microscope. Simultaneously, cells were collected at each time-point. The relative expression of the NNV main capsid protein (MCP) gene was assumed to reflect the virus level and *β-actin* was selected as the reference gene. Primers are listed in Appendix A. The mRNA expression of inflammatory factors and apoptosis-related genes, including tumor necrosis factor-alpha (TNF-α), interleukin 1-beta (IL-1β), IL-6, IL-8, interferon-gamma (IFN-γ), FADD, p53, Bcl-2, Bax, Caspase 3, Caspase 6, and Caspase 8, in CAB cells were determined by qRT-PCR. *EF-α* was selected as the reference gene.

For the purpose of examining the miRNAs expression changes in CAB cells post NNV infection, we selected the 5 known DEmiRs (miR-132-3p, miR-194a, miR-155, miR-203b-5p, and miR-146) which were closely related to innate immunity based on previous research reports and examined their expression after CAB cell infection with NNV by qRT-PCR method.

### 2.8. Immunofluorescence Microscopy Assay (IFA)

CAB cells were infected with NNV as described above. After 24 h post-infection, the cells were washed with PBS twice, fixed with 4% (*v*/*v*) paraformaldehyde (PFA) and permeabilized with Triton X-100 (0.2% in PBS) for immune-fluorescent staining [35]. After washing three times with PBS and blocked with 5% bovine serum albumin (BSA) for 60 min at room temperature, the cells were incubated with anti-MCP polyclonal antibody prepared by us at 1/200 dilution in 0.2% BSA at 4 °C overnight. The cells were washed three times with PBS and incubated with Alexa 488-conjugated secondary antibody at 1/1000 dilution for 1 h at room temperature. After washing three times with PBS, the cells were covered with antifade mounting medium (including DAPI) (Coolaber, Beijing, China) for 10 min and observed under an inverted fluorescence microscope.

### 2.9. miRNA Mimic, Inhibitor and Their Transfection

We transfected CAB cells with miRNA mimic or inhibitor at different concentrations (50, 100, 200 nM) to determine the best dosage for overexpression or inhibition effect. Cells were transfected with mimic, cal-miR-155 inhibitor, or miRNA controls, using TransIntro^®^ EL Transfection Reagent (TransGen, Beijing, China) according to the manufacturer’s protocol. The sequences for miRNAs are shown in Appendix A. After 24 h post-transfection, the cells were infected with NNV as described in Section 2.7.

### 2.10. Apoptosis Assay by Fluorescence Microscope and Flow Cytometry

CAB cells were cultured and infected with NNV as described in Section 2.7. At the end of 24 hpi, apoptosis assays by fluorescence microscope were conducted referring to Sun’s method with some modifications [36]. Briefly, the cells were washed twice with PBS after removing the medium, and then incubated with binding solution, which contained two kinds of staining (Annexin V-FITC and PI), for 15 min at room temperature in the dark. Then, the cells were covered with antifade mounting medium (including DAPI) (Coolaber, Beijing, China) for 10 min and observed using an inverted fluorescence microscope. For flow cytometry assays, the CAB cells were digested and gently washed with PBS twice. Then, the cell suspensions were incubated with 100 µL 1 × binding buffer containing the two stains as above and examined using a Guava easyCyte™ Flow Cytometer (EMD Millipore Corp., Billerica, MA, USA).

### 2.11. Western Blot Assay

The CAB cells were digested, seeded in a Φ10 cm cell-culture dish and incubated at 26 °C. After a 24 h incubation and transfected with the cal-miR-155 mimic or cal-miR-155 inhibitor, cells were infected with NNV as described in Section 2.7. CAB cells from different groups were harvested using a cell lysis buffer (Beyotime, Shanghai, China) containing 1 mM phenylmethylsulfonyl fluoride (PMSF) and was added to 1 × SDS-PAGE loading buffer, boiled for 10 min and separated by 12% SDS-PAGE, followed by transferring to a nitrocellulose membrane (Millipore, Darmstadt, Germany). After blocking with 5% BSA for 2 h, the membranes were incubated with diluted primary polyclonal antibody (anti-MCP, 1/1000 dilution) at 4 °C overnight and secondary antibody (HRP-conjugated goat anti-mouse IgG, 1/2000 dilution) for 1 h at room temperature after washing in TBST for five times of 5 min, respectively. Anti-β-tubulin mouse monoclonal antibody (Bioss, Beijing, China) was used as an internal reference. The results were detected with super sensitive ECL substrate (Biosharp, Anhui, China).

### 2.12. Analysis of Heat Shock Protein (HSP)70 by Flow Cytometry

Heat shock proteins (HSPs) are a group of highly conserved proteins that are expressed in every living organism. The HSP70 expression in CAB cells was measured by flow cytometry as previously described with slight modifications [37,38,39]. Briefly, the CAB cells, seeded in a Φ10 cm cell-culture dish, were suspended using 0.25% trypsin (Gibco). After PBS with 0.5% BSA washing, the cells were fixed with 4% PFA at room temperature for 10 min and then permeabilized with 90% ice-cold methanol for 30 min on ice. The cells were incubated with 10% goat serum for 30 min at room temperature to block non-specific protein-protein interaction. Subsequently, 1:100-diluted rabbit anti-HSP70 polyclonal antibody (BIOSS, Beijing, China) was added and incubated at room temperature for 30 min (Rabbit IgG were used as isotype control). After washing, the cells were incubated with the Alexa-488 labeled anti-rabbit antiserum at room temperature for 45 min. After washing twice, the cells were resuspended in PBS and analyzed using flow cytometry (CytoFLEX, Beckman, MIA, CA, USA). As a negative control, a mock treatment group that had no interaction with primary antibodies was incubated with Alexa-488 labeled secondary antibodies.

### 2.13. Statistical Analysis 

Statistical analysis was performed using GraphPad Prism (version 8.0.2) (GraphPad Software, San Diego, CA, USA). Statistical significance was evaluated in one way analysis of variance followed by Tukey’s multiple comparison test. Data were expressed as mean ± standard deviation (SD).

## 3. Results

### 3.1. Summary of the High-Throughput miRNA Data

In order to explore the miRNAs involved in NNV infection of *C. altivelis* spleen, the NNV infected *C. altivelis* spleen miRNA libraries from the third and the eighth day, as well as the PBS control group, were constructed and sequenced, named 3D, 8D, and control, respectively. Before the high-throughput sequencing was conducted, we performed a pre-experiment. The fish were infected by NNV supernatant and PBS (as a control) for 14 days. On the third day post-infection fish showed inappetence in the infected group. Several infected fish showed abnormal swimming behavior and began to die on the eighth day post-infection. Nevertheless, fish in the control group showed intact epithelial surfaces and no death at each time point. The UV treatment inactivated the NNV suspension, and was proved non-toxic to fish (data not shown).

After sequencing, 11,437,819, 11,642,611, and 9,526,201 reads of raw data were obtained from control, 3D, and 8D libraries. After filtering, 10,142,275 (88.67% of raw data), 9,592,236 (82.39% of raw data), 7,860,736 (88.67% of raw data) were drawn out as high-grade clean reads form the three sample groups, respectively (Appendix A). All of the clean data were with a similar Q30 of over 97%. After merging the same read sequences in clean reads, 332,239, 436,517, and 300,466 unique reads were obtained (Appendix A), with the majority of length 22 nt (Appendix A), a common miRNAs size. According to the unique reads count, the maximum number of unique sequence lengths from 17 nt to 35 nt were all found in the 3D library, compared to the other two libraries. In order to check whether the samples were contaminated, Bowtie analysis was used to map the reads to the reference genome of *C. altivelis*, the reads which did not match were filtered out. At last, the percentage of the clean reads count with the perfect match to the reference were all above 97%, and that for the unique reads was above 76%. Bowtie analysis was also used to align reads with exon and intron sequences of the reference genome, reads only on exons were filtered out. Including the known and the novel, the total number of unique miRNAs that we identified from the control, 3D and 8D libraries were 5,669,967 (73.12%), 7,249,296 (69.82%) and 6,207,487 (73.94%), respectively. Finally, we obtained 30,405, 29,548, and 28,108 miRNAs from the 3D, 8D, and control libraries (Appendix A). The 20 most numerous miRNAs constituted 84.87%, 86.25%, and 83.80% of the total miRNA counts in the control, 3D, and 8D groups, respectively (Table 1). As a result, 676 novel miRNAs could form hairpin structures and representative candidates are listed in Table 2. 

### 3.2. Phylogenetic Conservation Analysis 

The microRNA family is a group of miRNAs from the same ancestor, usually with similar biological functions, but do not necessarily have conserved miRNA primary and secondary structures. Among the clean reads, we detected 6955 miRNAs in the miRBase, 96.38% of them (6703 miRNAs) belonged to 197 miRNA families. Among them, let-7 was the largest miRNA family which has 13 paralogs. The top 10 families included miR-10, miR-17, miR-30, miR-25, miR-8, miR-15, miR-181, miR-29, and miR-130. The 6703 miRNAs mapped to 456 kinds of miRNAs. In order to investigate the conservation of the miRNAs we obtained among different species, the clean reads were mapped onto 42 vertebrate species with miRBase 22.1 using the miRDeep2 software, covering the Mammalia, Aves, Reptilia, Amphibia, Osteichthyes, and Cyclostomata. Finally, we identified 5814 miRNA matures and 1312 were matched to the eight bony fish miRNAs from miRBase (Appendix A). We analyzed the evolutionary conservation of our miRNA families to the animal kingdom after we sorted the 100 conserved miRNA families into nine groups base on species evolution (Appendix A).

### 3.3. Analysis of Differential Expression miRNAs

In total, 32 miRNAs were differentially expressed in two infected groups compared to the control group. In the 3D group, there were 27 miRNAs that were significantly differentially expressed, while only seven in the 8D group. Two miRNAs (cal-novel-441 and cal-novel-607) were shared between the 3D and 8D groups (Figure 1A). Then, we drew volcanic and heat maps between the infection groups and control group to speculate and explore the expression patterns of the DEmiRs during the infection process (Figure 1B,C). As shown in the volcano maps, among the 27 DEmiRs in the 3D group, 12 were up-regulated and 15 were down-regulated, including nine known miRNAs (cal-miR-132-3p, cal-miR-146b, cal-miR-155, cal-miR-184, cal-miR-203a-3p, cal-miR-203b-5p, cal-miR-206-3p, cal-miR-2188-3p, cal-miR-9-5p) and 18 novel miRNAs (cal-novel-287, cal-novel-274, cal-novel-276, cal-novel-110, cal-novel-14-star, cal-novel-144, cal-novel-221, cal-novel-229, cal-novel-23, cal-novel-26, cal-novel-28, cal-novel-30, cal-novel-33, cal-novel-34, cal-novel-41-star, cal-novel-441, cal-novel-607, cal-novel-8-star). Of the seven DEmiRs in the 8D group, five were up-regulated and two down-regulated. Among them, there was one known miRNAs (cal-miR-194a) and six novel miRNAs (cal-novel-401, cal-novel-435, cal-novel-441, cal-novel-46-star, cal-novel-529, cal-novel-607).

### 3.4. GO and KEGG Analyses of the Predicted DEmiRs Target Genes

To better characterize the function of the *C. altivelis* miRNAs, miRanda (v3.3a) was used to identify the target genes. The seed sequences, which always begin at the 2-8 nt sequences starting from the 5′ end of miRNA, were selected and predicted with the 3′-UTR of the target mRNAs. Then, the results showed that 1093 miRNAs predicted 16,513 target genes. Fifty-two and one target genes were predicted from 3D and 8D known DEmiRs, and 804 and 38 were from their novel DEmiRs, respectively. As we know, cluster analysis of significantly differentially expressed genes can effectively find the common expression patterns of different genes and infer the similarity of gene function according to the similarity of expression. As showed in Figure 2, after we subjected these target genes to analyze GO and KEGG pathway, we finally clustered 3099 and 306 GO terms for the 3D and 8D groups.

### 3.5. Verification of miRNA-Seq Results

Fourteen randomly selected miRNAs (five known DEmiRs and nine novel DEmiRs) were analyzed by qRT-PCR to confirm the sequencing quality. From the results shown in Figure 3, the expression of these 14 miRNAs were in accordance with those from sequencing. This meant our deep sequencing results were credible.

### 3.6. CAB Cell Infected by NNV

In order to explore the function of host miRNAs in cell level, the sensitivity to NNV of the CAB cell line was determined. The results show that the CAB cells infected with NNV exhibited obvious CPEs and multiple vacuolations at 24, 48, and 72 hpi compared with the PBS control and UV-inactivated NNV infected cells (Figure 4A). Meanwhile, qRT-PCR of the NNV MCP gene was performed to evaluate the NNV replication in CAB cells. As demonstrated in Figure 4B, the relative expression of the MCP grew in a time-dependent manner in the NNV-infected group. In contrast, no obvious CPEs were observed in UV-inactivated NNV-infected cells, and the MCP mRNA expression was comparable to the cells of the PBS control. These results prove that the tissue grinding fluid was not the cause of the cell cytopathic effect and indicates that NNV replicated well in CAB cells. As shown in Figure 4C, strong fluorescence signals were observed in infected cells, but not in the control group. This indicated that the antibody prepared for NNV mcp reacted specifically with intracellular viral particles, and also substantiated that the NNV virus was proliferating well in the CAB cells. Previous studies have shown that apoptotic signaling is often consistent with observations of high levels of HSPs [40]. Reports have shown that HSP70 is highly conserved and has antiapoptotic activity [38,41,42]. Thus, in the present study, HSP70 was detected as an apoptotic marker by flow cytometry. The fluorescence histogram reveals the percentage of HSP70 expression level in CAB cells. It indicates that the expression of HSP70 was remarkably up-regulated in the NNV-infected cells (Figure 4D). Subsequent results of double staining assays by Annexin V/PI indicate that the CAB cells exhibited both green and red signals after infection with NNV (Figure 4D). These results suggest that NNV infection caused apoptosis in the CAB cells. Then, we further determined the impact of viral stimulation on mRNA expression of inflammatory factor genes, including tumor necrosis factor-alpha (TNF-α), interleukin 1-beta (IL-1β), IL-6, IL-8, interferon-gamma (IFN-γ), and apoptosis-related genes, including FADD, p53, Bcl-2, Bax, Caspase 3, Caspase 6, and Caspase 8, in CAB cells by qRT-PCR. The results indicate that the cells infected with NNV showed significant up-regulation of most of these genes, except Bax and TNF-α, compared with the controls (Figure 4F). Taken together, NNV infection could induce cellular apoptosis and inflammatory responses in CAB cells.

### 3.7. Infection of CAB Cells with NNV Increased cal-miR-155 Expression

The expression of five known DEmiRs were examined after infection with NNV in the CAB cells. Among them, cal-miR-194a and cal-miR-155 expression at 24 h increased by 2.22-fold, and 5.78-fold, respectively, after NNV infection (Figure 5A). Therefore, we decided to investigate the role of cal-miR-155 in the host response to NNV infection. Subsequently, CAB cells were infected with NNV and the expression of cal-miR-155 at 0, 24, 48, and 72 hpi was examined. From the results in Figure 5B, cal-miR-155 was markedly up-regulated at 24 hpi, but as the time went by, the relative fold expression of cal-miR-155 decreased. These results are illustrative that cal-miR-155 might make an important contribution in host cell response to NNV infection.

### 3.8. Cal-miR-155 Enhances Cell Apoptosis in CAB Cells

It has been previously found that NNV infection can induce cell apoptosis and increase the expression levels of cal-miR-155. Thus, we sought to investigate if there was any relationship between cal-miR-155 expression and cell apoptosis. After transfection, cal-miR-155 mimic at 100 nM and 200 nM greatly increased the expression of miR-155, but the 100 nM showed higher up-regulated effects (Figure 6A). For the cal-miR-155 inhibitor, it showed an obvious suppressive effect in cal-miR-155 expression at 100 nM and 200 nM; but unlike the mimic, it was at 200 nM concentration that showed best inhibition effect (Figure 6B). Whereafter, flow cytometry was used to check apoptosis at 24 h post-transfection. The cells transfected with cal-miR-155 presented a greater apoptotic rate than those transfected with mimic controls (Figure 6C above), while the cells transfected with the cal-miR-155 inhibitor exhibited significantly inhibited apoptosis. Subsequently, the role of cal-miR-155 in NNV-induced apoptosis attracted our attention. The results in Figure 6C (lower) display that cal-miR-155 could obviously enhance the apoptotic effect in CAB cells which already begin apoptosis after infection with NNV. When its expression was suppressed by the cal-miR-155 inhibitor, the apoptosis effect in NNV-infected CAB cells was reduced. Flow cytometry was also used to further analyze HSP70 expression, which is considered to be a reflection of the cells apoptotic level. According to the result in Section 3.6, NNV infection can obviously increase HSP70 expression. After overexpression of cal-miR-155, the level of HSP70 was up-regulated, while its expression was down-regulated after cells were transfected with the cal-miR-155 inhibitor with or without NNV infection (Figure 6E). These results are consistent with the apoptotic trend previously observed. Taken together, the results demonstrate that cal-miR-155 could enhance cell apoptosis whether there is virus present or not.

### 3.9. miR-155 Effects the mRNA Expression of Apoptosis-Related Genes in CAB Cells

Then, we investigated the contribution of cal-miR-155 on regulating apoptosis-related genes in CAB cells. We transfected cal-miR-155 mimic and inhibitor into CAB cells. After 24 h, the cells were harvested, and apoptosis-related genes were detected by qRT-PCR. As shown in Figure 7A, the cal-miR-155 mimic significantly up-regulated the expression of FADD, p53, Bax, and Caspase-3/6/8, compared with the mimic control NCs. While the mRNA expression of the gene Bcl-2, an anti-apoptotic gene, was significantly down-regulated. On the contrary, the cal-miR-155 inhibitor significantly increased the expression of Bcl-2 and caspase-3. While, after transfection at 24 h, the CAB cells were infected with NNV. After 24 hpi, the apoptosis-related genes mRNA expressions showed that (Figure 7B), the cal-miR-155 mimic significant up-regulated the expression of FADD and Bax, compared with the mimic control NCs. While Bcl-2 was still significantly down-regulated. On the contrary, the cal-miR-155 inhibitor significantly increased its expression.

### 3.10. cal-miR-155 Inhibits NNV Replication in CAB Cells

Since cal-miR-155 has been proved to have the ability to promote cell apoptosis induced by NNV, we tried to investigate its impact on NNV replication. We transfected CAB cells to overexpress a cal-miR-155 mimic in vitro and examined the relative expression of the *MCP* gene at 24, 48, and 72 hpi. As shown in Figure 8A, MCP mRNA expression significantly decreased in NNV-infected cells, suggesting that overexpression of cal-miR-155 could inhibit NNV replication. After we inhibited the expression of cal-miR-155 in CAB cells with a cal-miRNA-155 inhibitor, the MCP gene expression in NNV-infected cells was up-regulated at 72 hpi (Figure 8B). We also evaluated the effect of miR-155 on the NNV mcp protein level, and the results were consistent with the qRT-PCR results. The NNV mcp protein level was lower in the miR-155 mimic transfected cells than that of the controls (Figure 8C). On the contrary, the mcp protein level was higher in the miR-155 inhibitor transfected cells than of the control group (Figure 8D). These data clearly show that cal-miR-155 plays an important role in suppressing NNV replication.

## 4. Discussion

In the past few years, high-throughput sequencing of miRNA responding to viral infection in teleost fish or fish cells and our knowledge on the function of these miRNAs has increased rapidly [43]. However, the molecular mechanisms underlying NNV infected *C. altivelis* is still not clear. In this study, first, the miRNAs caused by NNV in *C. altivelis* spleen were identified, miRNA libraries of CAB cells infected with NNV at early (3 dpi) and late (8 dpi) stages were constructed, and many DEmiRs were obtained. Secondly and importantly, one of the DEmiRs, cal-miR-155, was screened out to show significant responses to the virus in CAB cells and its overexpression proved to induce apoptosis. As a result, the replication of virus was suppressed. All these results demonstrate that miRNAs may have a critical role in defending against NNV infection.

Up to now, people have completed high-throughput miRNA sequencings on teleost fish and cells in response to various viral infections, such as grouper to Singapore grouper iridovirus (SGIV) and NNV, flounder to megalocytivirus and VHSV, grouper fin (GF-1) cells to VHSV, and epithelioma papulosum cyprinid (EPC) cells to VHSV [11,12,13,14,15]. In compliance with studies in other species, for example, *P. olivaceus* and *Cynoglossus semibreves,* we also found the majority of miRNA lengths was 22 nt according to the length distribution, a typical length of most animal miRNAs [44,45]. We also classified the 100 miRNA families into nine groups base on the species evolution time, and then analyzed the evolutionary conservation of our miRNA families to the animal kingdom. The results suggest that the miRNA family of *C. altivelis* is highly conserved among different animal species.

According to a lot of previous studies, plenty of DEmiRs have been found during the virus infection process and the expressions of host miRNAs could be affected during this period [12,14,15]. In our study, twenty-seven and seven DEmiRs were found from 3D and 8D libraries, respectively. Obviously, the number of DEmiRs showed a declining trend. GF-1 cells displayed a similar trend with 51 and 16 DEmiRs in the early (3 hpi) and late (24 hpi) stages after infection with the RGNNV, respectively [14]. While it was not all the same, compared to the number at 6 and 12 hpi, the number of DEmiR in VHSV infection olive flounder greatly increased at 24 hpi [13]. This may be owing to the different viral infection systems. In the current study, the DEmiRs exhibited different expression patterns. At early stages, 16 DEmiRs were up-regulated, while at late stages, the number was 14, compared to the control. Most of the DEmiRs that were found in this study were assumed to have important function in the antiviral processes. For example, in mammals, researchers revealed that miR-132-3p overexpression could promote the replication of the influenza A virus (IAV) [46]. High-levels of miR-146a facilitated SGIV replication and reinfection through retarding virus-induced apoptosis in groupers [47]. miR-155 was also exhibited anti-CyHV-3 activity in common carp brain (CCB) cells [48]. Since majority of miRNAs actthrough binding to the target genes to exert their function, the prediction and analysis of miRNA target genes is very important in understanding the function of miRNAs [49]. In this study, ten known DEmiRs and 22 novel DEmiRs targeted 53 and 842 genes of *Cromileptes altivelis*, respectively.

It is commonly known that NNV exerts its effects mainly at the central nervous system [50]. Therefore, fish brain cell lines may become a useful tool for viral study. Some fish brain cell lines have been established for NNV pathogenesis studies, such as *Lates calcarifer* and *Lateolabrax japonicus* [51,52]. According to our previous study, the CAB cell line is susceptive to *Vibrio harveyi* and *Edwardsiella tarda* [21]. In the current study, we observed typical CPEs after NNV-infection, and the virus replication was also confirmed by qRT-PCR. These indicated that we could use the CAB cell line to study the interactions of host and NNV in vitro. On this basis, we tried to investigate the function of miRNAs in the CAB cell response to NNV infection. Among the five known DEmiRs (miR-132-3p, miR-194a, miR-155, miR-203b-5p, and miR-146), cal-miR-155 showed the highest up-regulation compared to the other four miRNAs, indicating that it might play an essential role in defending against NNV infection in host cell lines. Previous studies have confirmed that miR-155 is related to the immune reaction and has antiviral effects, such as regulating cytokine expression of *Cyprinus carpi*, and exhibiting anti-CyHV-3 activity in common carp brain (CCB) [48,53,54]. According to the results after cell infection at different time points, the expression of cal-miR-155 markedly decreased after 24 hpi, indicating that the function of cal-miR-155 might act in early-stages of the NNV infection. This is consistent with our findings that cal-miR-155 increased at 3 dpi but decreased at 8 dpi, indicating that it might respond to NNV infection in the early stage.

As we mentioned in front, NNV is a non-enveloped virus and needs apoptosis of the host cells to occur for its release. Like most viruses, NNV infection can induce apoptosis [55,56]. Similar findings also appear in the avian reovirus (ARV), another non-enveloped virus [57]. In this study, we confirmed that NNV infection leads to CAB cell apoptosis by Annexin V-FITC/PI double staining and is associated with the increased expression of cal-miR-155. Therefore, we speculated whether cal-miR-155 had any effects on cell apoptosis. To verify this hypothesis, we transfected a mimic or inhibitor to overexpression or suppress cal-miR-155 levels in CAB cells. Results displayed that transfection of the cal-miR-155 mimic significantly increased the apoptosis and, conversely, inhibited cal-miR-155 decreased apoptosis with or without NNV presence. Numerous studies have shown that HSPs are not only heat shock-inducible proteins, but can also be stimulated by other factors, including growth factors, infections and inflammation [58]. During the course of viral infection, the expression of heat shock genes are induced by activation of the cellular stress response and play a key role in regulating apoptosis [40,58]. In this study, NNV infection steeply induced HSP70 expression in CAB cells. This suggested that apoptosis caused by viral infection was accompanied with the HSP70 increase. Moreover, it was also found that transfecting the cal-miR-155 mimic increased HSP70 expression levels significantly with or without the presence of NNV. In the following experiemnts, the opposite result was found after transfection with the inhibitor. It is generally accepted that apoptosis consists of two classic pathways: the intrinsic pathway (mitochondrial pathway) and the extrinsic pathway (receptor-mediated pathway) [59,60]. In our research, the mRNA expression of FADD and TNF-α which belong to extrinsic pathway were not significantly altered after NNV infection, indicating that it was the intrinsic pathway that was activated in CAB cells post-infection. At this time, the cells need to promptly inhibit the occurrence of apoptosis to survive under the pathogen stimulus. This is consistent with our results that the anti-apoptotic factor *Bcl-2* was significant increased, while the pro-apoptotic factor *Bax* was significant decreased. However, the expressions of p53 and caspase 3/6/8 were up-regulated stimulated by the virus, accompanied by the up-regulation of several inflammatory factors. These actions are common to reduce apoptosis induced by viral stimulation. Recently, it was reported that during ISKNV infection in GF-1 cells, Bcl-2/Bcl-xL interacts with Bax/Bak in a dynamic interaction. This is to maintain the mitochondrial function in GF-1 cells [61]. In the current study, transfection of the cal-miR-155 mimic significantly increased apoptosis with or without NNV presence. If the purpose of virus-promoted apoptosis is to release virions and infect more cells, we hypothesize that the mechanism of promoting apoptosis may be a process of suicidal self-protection. According to studies on mammals, miR-155 also has two roles in the regulation of apoptosis. In human THP-1 cells infected with Bacillus Calmette-Guerin (BCG), miR-155 is up-regulated to inhibit cell apoptosis and causes bacilli to escape the host and avoid infection [62]. On the contrary, overexpression of miR-155 enhances apoptosis in a time-dependent manner, suggesting that miR-155 promotes apoptosis, which is consistent with our findings [63]. We have previously discussed that cells activate the intrinsic apoptosis pathway to inhibit their own apoptosis under viral stimulation. Therefore, it perplexes us that apoptosis induced by cal-miR-15 is an enhanced response, or is this mediated through another pathway? For this purpose, *FADD*, *p53*, *Bcl-2*, *Bax*, *Caspase-3/6/8* were detected in both cells infected with and without NNV after transfection with a cal-miRNA-155 mimic or inhibitor for 24 h. Before NNV stimulation to the cell, the expressions were all up-regulated upon cal-miR-155 overexpression, except *Bcl-2*. *FDAA* and *Caspase-3* were down-regulated, and simultaneously, *Bcl-2* was up-regulated, when expression of cal-miR-155 was inhibited. Consistent with this result, AMO-155 (antisense inhibitor) has been reported to reverse H_2_O_2_-induced down-regulation of Bcl-2, up-regulation of Bax and cleaved caspase-3 in mice [64]. gga-miR-16-5p has been reported toinhibit the expression of Bcl-2 and enhance FAdV-4 (fowl adenovirus serotype 4)-induced apoptosis [65]. In combination with what was discussed earlier, cal-miR-155 may activate the extrinsic receptor-mediated pathway but this still needs further experiments to prove.

Apoptosis induced by viral infection has both positive and negative effects on virus replication [59]. After NNV stimulation, the cells transfected the cal-miR-155 mimic showed the highest degree of apoptosis, this may be the result of the combined action of the two apoptotic pathways. Subsequently, we found that the enhancement of cal-miR-155 had on apoptosis in NNV-infected cells suppressed viral replication. Exactly how miR-155 promotes apoptosis to inhibit the proliferation of the virus and what the target genes are still needs further screening and experimental verification.

In conclusion, the current study provided an miRNA transcriptome analysis upon NNV infection of *C. altivelis* and showed that NNV infection changes the host miRNAs expression in vivo and in vitro. In CAB cells, we found that the expression of cal-miR-155 was up-regulated by NNV infection. cal-miR-155 increased NNV-induced apoptosis and resulted in inhibiting NNV replication. We speculated that NNV induces cell apoptosis through the intrinsic pathway, while cal-miR-155 could increase the apoptotic effect through the extrinsic pathway. From all of these results, cal-miR-155 may contribute in the anti-viral process in host cells and provide new insights in understanding the function of host-virus interaction.

## Figures and Tables

**Figure 1 viruses-14-02184-f001:**
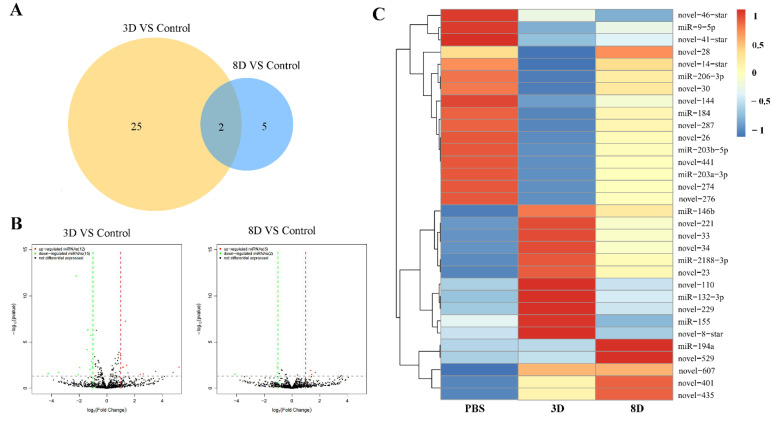
Analysis of differentially expressed miRNAs (DEmiRs) in *C. altivelis* following NVV challenge. (**A**) Venn diagram shows DEmiRs numbers in the two infected groups compared to the control group. The DEmiRs number between the 3D and control groups was orange, while it was light blue between the 8D and control groups. (**B**) The volcano map shows the overall distribution of DEmiRs. Red color represents significantly differentially up-regulated, while the green color represents significantly differentially down-regulated. Black dots represent non-significantly differential expression. (**C**) The expression patterns for the 32 DEmiRs. The expression levels of these 32 miRNAs are shown in different colors. The red color indicates the highly expressed DEmiRs, and the blue color for the lowly expressed DEmiRs.

**Figure 2 viruses-14-02184-f002:**
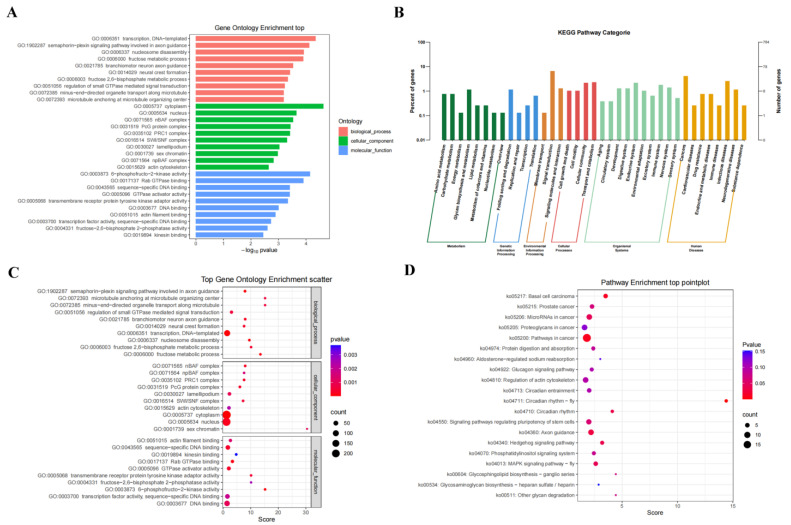
GO and KEGG pathway enrichment analysis of the target genes of DEmiRNAs from the spleen in *C. altivelis* following NVV challenge. The DEmiRNAs caused by NNV infection were matched to various GO and KEGG pathway categories. (**A**) Bar chart of top10 statistics of GO enrichment of DEmiRNAs target genes in biological process, cellular component, and molecular function, respectively. (**B**) Bar chart of KEGG enrichment of DEmiRNAs target genes. (**C**) Significant GO enrichment function scatters plot of DEmiRNAs target genes. (**D**) Scatter plot of KEGG pathway enrichment top20 statistics after NNV infection.

**Figure 3 viruses-14-02184-f003:**
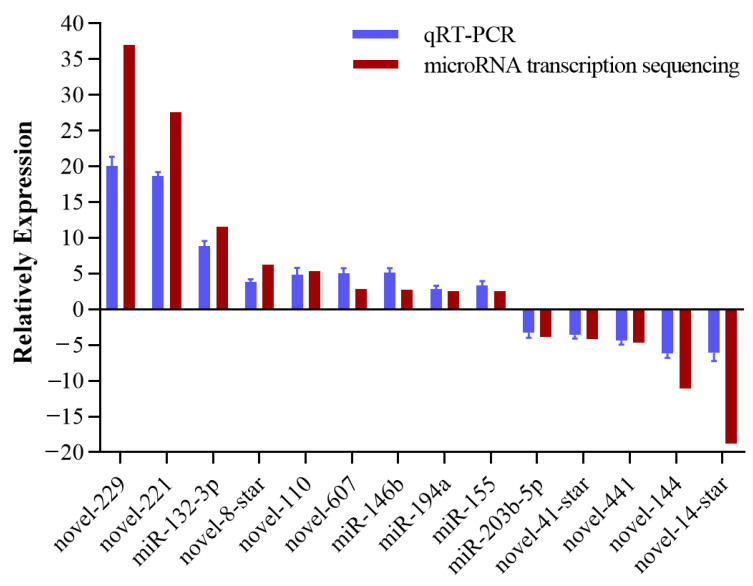
Validation of miRNA sequence data by qRT-PCR. Fourteen miRNAs were selected to validate the miRNA-seq data using qRT-PCR. Relative expression levels of change between the infected and control groups were calculated and normalized by the U6 housekeeping. Data in this assay were expressed as mean ± SEM (*n* = 3).

**Figure 4 viruses-14-02184-f004:**
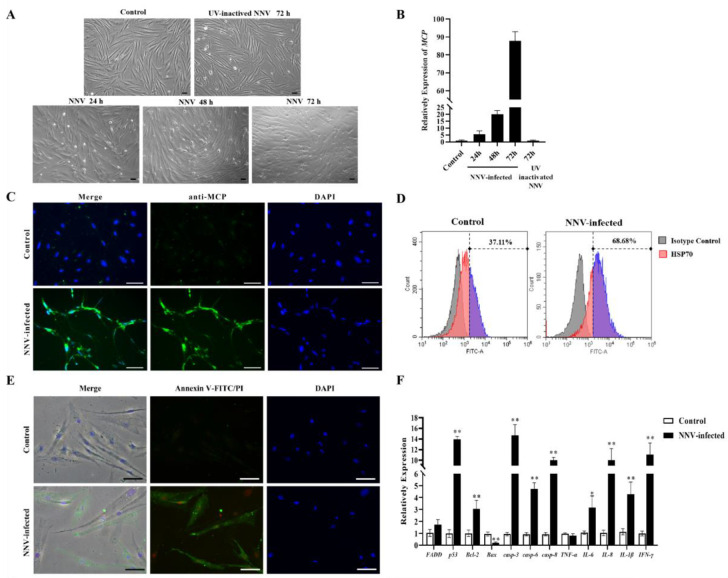
CAB cells infected with NNV. (**A**) CPE of CAB cells infected with NNV (MOI = 10) were observed at 24, 48, and 72 hpi, respectively, compared with PBS control and UV-inactivated NNV infected cells. (**B**) Analysis of *MCP* mRNA expression in NNV-infected CAB cells at 24, 48, and 72 hpi, compared with PBS control and UV-inactivated NNV at 72 h. (**C**) NNV replication in cells detected by immunofluorescence microscopy. The upper row shows uninfected CAB cells (control), whereno fluorescence staining was observed; the lower row shows NNV-infected CAB cells, in which strong fluorescence signal was observed in the virus-infected cells. (**D**) Flow cytometric analysis of HSP70 expression in CAB cells infected with or without NNV at 24 h. Fluorescence histogram illustrates HSP70 expression, and the gate was applied using an isotype control. Anti-HSP70 antibody and Alexa-488 labeled secondary antibody were used for probing. The color purple represented the relatively expression level of HSP70. (**E**) Cell apoptosis was detected using annexin V-FITC/PI staining in CAB cells infected with PBS (upper) and NNV (lower) after 24 h. Green and red fluorescence represents early- and late-stage apoptosis, respectively. Scale bar = 50 μm. (**F**) The mRNA expression of inflammatory factor genes and apoptosis-related genes in CAB cells infected with NNV. The CAB cells infected with NNV at 24 h were collected and the total RNA was extracted. Expression levels of proinflammatory cytokines and apoptosis-related genes were determined by qRT-PCR, with the expression level of each gene of the PBS control group as 1. *EF-α* was selected as the reference gene. Data in this assay is expressed as mean ± SD (*n* = 3). *n*, the number of cell wells used in the experiment. (** stands for *p* < 0.01; * stands for *p* < 0.05.).

**Figure 5 viruses-14-02184-f005:**
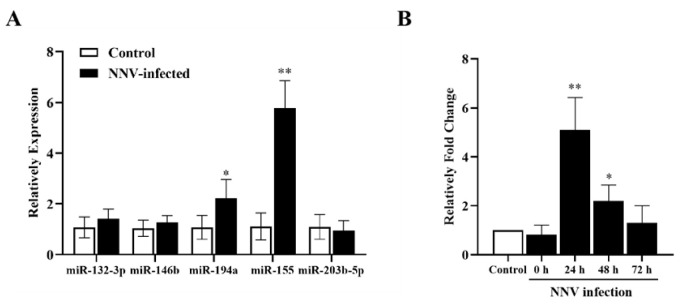
Infection of CAB cells with NNV increased cal-miR-155 expression. (**A**) The expression of five known DEmiRNAs were examined after CAB cells were infected with NNV (MOI = 10) after 24 h. (**B**) CAB cells were infected with NNV or PBS as a control. The cells were collected at different time points after infection to extract total RNA with TRIzol and amplified by reverse transcription. The expression of cal-miR-155 in response to NNV infection in CAB cells at 24, 48, and 72 hpi was examined by qRT-PCR. The relative fold change in cal-miR-155 expression in NNV-infected CAB cells was compared to that in PBS-infected cells control. Asterisks indicate significant differences between 0, 24, 48, or 72 hpi with the PBS-infected cells. U6 was used as an internal control. (** stands for *p* < 0.01; * stands for *p* < 0.05.).

**Figure 6 viruses-14-02184-f006:**
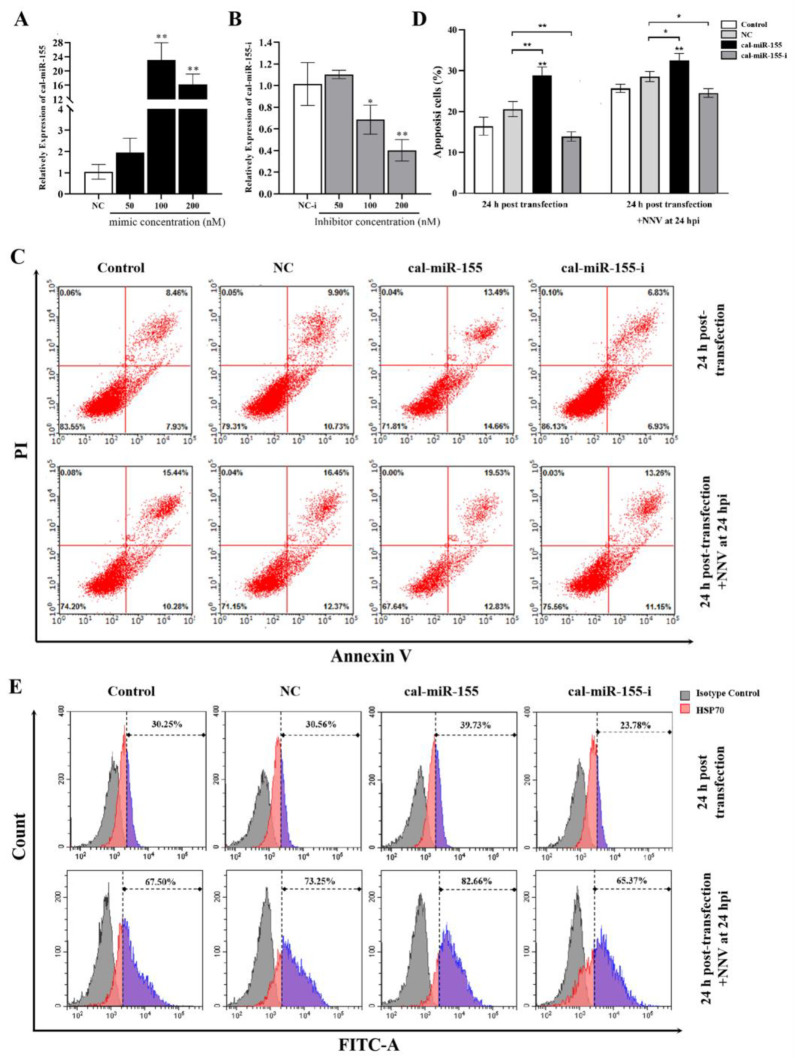
Cal-miR-155 enhances apoptosis in CAB cells with or without NNV infection. (**A**) The relative mRNA expression level of cal-miR-155 in CAB cells transfected with 50 nM, 100 nM and 200 nM cal-miR-155 mimic (miR-155) compared to that of cells transfected by cal-miR-155 mimic control (NC); detected by qRT-PCR after 24 h post-transfection and normalized to U6. (**B**) The relative mRNA expression level of cal-miR-155 in CAB cells transfected with 50 nM, 100 nM and 200 nM cal-miR-155 inhibitor (miR-155-i) compared to that of cells transfected by a cal-miR-155 inhibitor control (NC-i); detected by qRT-PCR after 24 h post-transfection and normalized to U6. (**C**) CAB cells transfected with miR-155, miR-155-i, or NC infected with or without NNV for 24 h, while untransfected CAB cells were used as control. Then the cells were harvested and stained with Annexin V-PE and PI, and the cell apoptosis was examined by flow cytometry. (**D**) The percentage of apoptotic cells in each group shown in panel C was graphed and statistically analyzed. Data are representative of three independent experiments and presented as means ± SD. (**E**) The HSP70 expression for those cells was measured by flow cytometry, and the gate was applied using an isotype control. The color purple represented the relatively expression level of HSP70. (** stands for *p* < 0.01; * stands for *p* < 0.05.).

**Figure 7 viruses-14-02184-f007:**
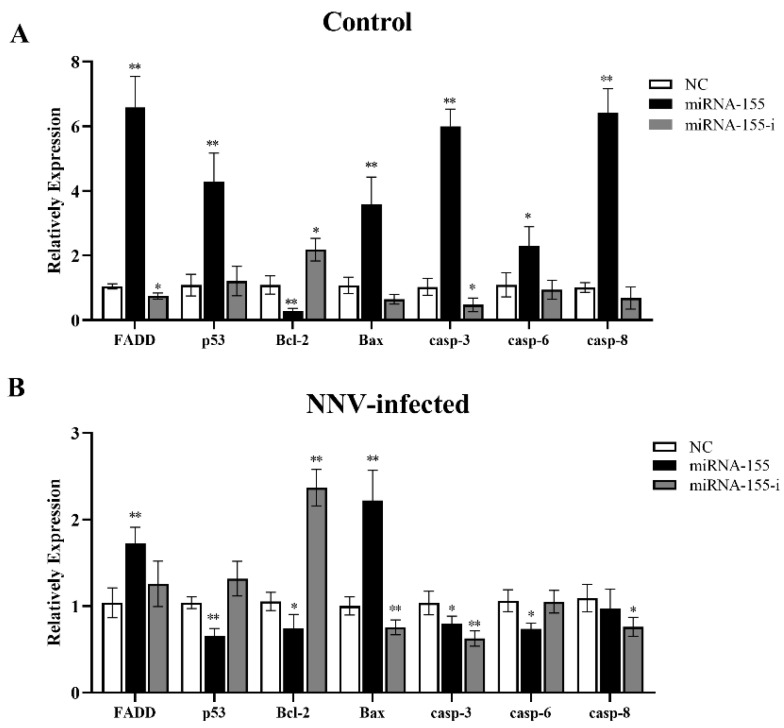
cal-miR-155 affects the mRNA expression of apoptosis-related genes in CAB cells. CAB cells were transfected with a cal-miR-155 mimic and an inhibitor for 24 h, then the cells were infected with NNV (MOI = 10), as described. After 24 h, the cells were harvested, and apoptosis-related genes, FADD, p53, Bcl-2, Bax, and Caspase-3/6/8, were detected in cells infected with NNV (**B**) or not (**A**). *EF-α* was selected as the reference gene. Data in this assay are expressed as mean ± SD (*n* = 3). (** stands for *p* < 0.01; * stands for *p* < 0.05.).

**Figure 8 viruses-14-02184-f008:**
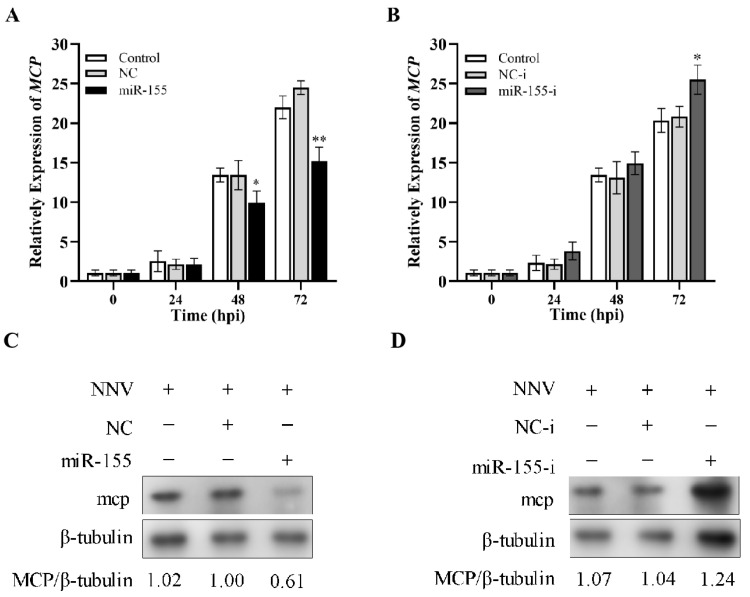
Cal-miR-155 inhibits NNV replication in CAB cells. After CAB cells were transfected with the cal-miR-155 mimic (**A**) and inhibitor (**B**) for 24 h, they were infected with NNV (MOI = 10). Then, the cells were harvested and the expression of the NNV *MCP* gene was detected at different time points (24, 48, and 72 hpi) by qRT-PCR. β-actin was selected as the reference gene. Data in this assay are expressed as mean ± SD (*n* = 3). (** stands for *p* < 0.01; * stands for *p* < 0.05.) CAB cells transfected with the cal-miR-155 mimic (miR-155) (**C**) or the cal-miR-155 inhibitor (miR-155-i) (**D**) for 24 h were infected with NNV at a MOI of 10. Then, the NNV mcp protein level was detected by western blot, and β-tubulin was used as the internal control. The gray values of NNV MCP/β-tubulin are shown on the last line.

**Table 1 viruses-14-02184-t001:** The 20 most abundant miRNAs of the three libraries in *C. altivelis* following NVV challenge. Expression profiles of miRNA were estimated by CPM (count per million).

PBS		3D		8D	
miRNA Name	Expression	miRNA Name	Expression	miRNA Name	Expression
cal-miR-144-3p	91,677.39	cal-miR-26a-5p	106,924.32	cal-miR-26a-5p	97,390.75
cal-miR-100-5p	88,587.59	cal-miR-146a	97,857.94	cal-miR-100-5p	90,016.18
cal-miR-26a-5p	88,509.13	cal-miR-144-3p	88,316.26	cal-miR-146a	82,117.31
cal-miR-146a	85,764.52	cal-miR-100-5p	68,439.57	cal-miR-144-3p	69,715.95
cal-miR-126a-3p	60,479.22	cal-miR-126a-3p	44,046.19	cal-miR-126a-3p	61,392.14
cal-miR-30e-5p	27,820.74	cal-miR-144-5p	30,394.52	cal-miR-144-5p	26,655.57
cal-novel-1-mature	24,203.63	cal-novel-1-mature	27,111.71	cal-miR-30e-5p	25,548.07
cal-miR-144-5p	23,185.2	cal-miR-30e-5p	25,524.39	cal-novel-1-mature	21,622.15
cal-miR-451	20,406.47	cal-miR-451	21,552.59	cal-miR-99	19,569.93
cal-miR-99	19,679.13	cal-miR-99	16,654.81	cal-miR-451	16,919.73
cal-miR-10a-5p	10,619.34	cal-miR-10a-5p	9711.92	cal-miR-10a-5p	9202.12
cal-miR-30d	8692.66	cal-miR-30d	9080	cal-miR-30d	9080.65
cal-let-7j	7445.7	cal-miR-142a-5p	6696.76	cal-let-7a	7960.16
cal-miR-125b-5p	7386.09	cal-novel-279-mature	6639.14	cal-miR-16b	7345.75
cal-let-7a	7377.74	cal-miR-181a-5p	6606.22	cal-let-7j	6946.54
cal-miR-181a-5p	7211.98	cal-let-7j	6382.04	cal-miR-181a-5p	6775.25
cal-miR-199-5p	6226.97	cal-miR-125b-5p	6297.61	cal-novel-279-mature	6709.32
cal-miR-142a-5p	6012	cal-let-7a	6283.08	cal-miR-125b-5p	6658.31
cal-novel-279-mature	5843.98	cal-miR-462	5703.18	cal-miR-199-5p	6122.14
cal-miR-16b	5783.36	cal-miR-199-5p	5630.75	cal-miR-30c-5p	5520.56
cal-miR-144-3p	91,677.39	cal-miR-26a-5p	106,924.32	cal-miR-26a-5p	97,390.75

**Table 2 viruses-14-02184-t002:** The precursor sequence, precursor coordinate on *C. altivelis* genome, and predicted hairpin structures of novel miRNAs.

miRNA	Consensus Precursor Sequence	Precursor Coordinate	Hairpin Structure	MFE (kcal/mol)
cal-novel-229	cuaguuugacaguuugaccgcaguucacuagcagugauugacaugacuaaca	Hic_asm_4:13483977..13484029:-	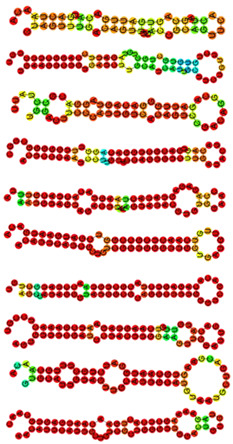	−8.10
cal-novel-221	ugaagucugugaucuugcaucauugcuucuguucugugugcugcuguagauuccaggcuucugu	Hic_asm_14:33758924..33758988:-		−19.40
cal-novel-529	uggacuucccauugucacaguccgacaggcagauugggacaguaggacgccuauu	Hic_asm_10:17820948..17821003:+		−15.00
cal-novel-8	acucccaauccuuguaccagugucuugauacuacagugacgcuggacagguuugggggcggc	Hic_asm_1:40300572..40300634:+		−30.70
cal-novel-110	aauguaguagacuuaaaguauacguguggaaccagagaguauacuaaguacacuacguuua	Hic_asm_18:26910902..26910963:-		−17.50
cal-novel-41	acauucaucgcugucgguggguuggugauguugucaacucgccggucgaugaaugaca	Hic_asm_1:38954803..38954861:+		−28.60
cal-novel-441	agccauaggguauggcgcaggcuuugggucgacacagagucuacgcuguaaccuaugcugu	Hic_asm_9:5259091..5259152:-		−28.90
cal-novel-144	aguguuucgauccuacuuggaaguuggauugacauguugucaaguagaaucgaaacuccgu	Hic_asm_0:44101038..44101099:+		−20.90
cal-novel-46	gugcauuguaguugcauugcauguaaugcugacgaagugcaauggaucugcuuugcaaca	Hic_asm_19:24959639..24959699:-		−20.10
cal-novel-14	aacauucauugcugucgcuggguuggacuguguagaaaagcucacugaacaaugagugcaac	Hic_asm_17:11344654..11344716:-		−16.60

## Data Availability

All data will be made available upon request from the corresponding author.

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
