# Peer review of "Cromileptes altivelis microRNA Transcriptome Analysis upon Nervous Necrosis Virus (NNV) Infection and the Effect of cal-miR-155 on Cells Apoptosis and Virus Replication"

_viruses, 2022, doi:10.3390/v14102184_

Round 1

Reviewer 1 Report

Summary:

In their manuscript, Du et al. suggest that NNV virus infection induce a wide range of miRNA including miR-155 which in turn activates multiple apoptosis-related genes and induce cell death. Although the study is interesting, multiple concerns arise including the lack of multiple controls. In addition, I would strongly suggest another round of English editing.

Major:

- Virus preparation: According to methods, virus was prepared from "diseased fish tissue homogenates". How pure is the pathogen solution. Did the authors checked for the presence of other pathogens? Virus should be ultracentrifuged or authors need to show using UV irradation and inactivation of the supernatant, that the innocculum is not toxic in itself.

- How are they sure NNV is the only virus there?

- There is a titration method providing "viral copy number" (line 82), but authors used TCID 50 as titer. The authors need to explain and stipulate how much virus was used for cell work exactly.

- Figure resolution is overall too low. It is hard to read them and some text is not readable at all.

- Section 3.1: What is the rationale for choosing day 3 and day 8 post-infection? A clear rationale should be stated. Also what is the status of the infection at days 3 and 8 post-infection. Authors need to show some controls of the infection.

- Figure 2 has multiple issues. Line 241: Where do these numbers come from? No mention in the previous 3.1 analysis section. How do they relate to the Venn Diagram? What does the Venn diagram (Fig2.a) represent? The figure legend is very poor.

- Figure 4: Figure legend suggest the experiment was only repeated once (albeit 3 biological repeats). Authors need 3 independent repeats. Why is there no statistical test perform? There is a typo: it should be sequencing and not "sequcecing". What cells were used? How long after infection?

-Figure 5: MCP mRNA expression is very minimal in comparison to day 0 which should only be noise. Especially in comparison to the cytopatic effect observed. they authors need to UV irradiate and inactivate and repeat the experiments to prove it is not related to the rest of the dead fish soup...

- Please show immuno-fluorescence of a marker for NNV infection for pannel Fig 5A/C.

- Figure 7C and 7D: There is some labelling mistake with NC-i missing despite mentioned in figure legend and no explaination for "normal" in figure legend. Please correct.

- Figure 8: Immunoblotting should be performed to confirm induction of all the apoptosis markers.

- Figure 9: Effect on NNV replication is far from convincing: fold change<<2. Please provide an additional measure of infection (Western blot, viral copy number or plaque assay).

Minor:

- Line 204 ("11,437,819, 11,642,611, and 9,526,201 raw data 204
were obtained
"): do you mean reads?

- Figure 1 could be transferred to the supplemental material section.

Reviewer 2 Report

Nodavirus is a non-enveloped single-stranded RNA virus, responsible for causing diseases from larvae to adult marine and freshwater fish species throughout the world. Thus, studies in the grade of increasing the knowledge about the immune system reaction against the virus and new drug/vaccination strategies to contrast the infections in culture are welcome. In this view, this Manuscript could have potentialities for the Viruses readers, however, it needs some further tests to confirm the findings (RT-PCR of miRNA expression in the head kidney of infected and uninfected fish; flow-cytometry with HSP70 antibody on infected and uninfected cells. A clear aim and focus of the research should be done (in the abstract, introduction and discussion sections). References should be implemented by some significant papers. 

Lines 14-15. “Humpback grouper (Cromileptes altivelis), a commercial fish of great importance for aquaculture”. Authors should insert “great importance for Asian aquaculture”.

Lines 16-18. The focus/aim and the link between the virus and the microRNA are not clearly explained. The authors should rewrite the sentence and expose with clarity why the miRNA expressions are important in contrasting a Virus infection and why it's fundamental to study the transcriptome focused on miRNA in infected/uninfected Fish.

 Lines 89-90: ”and the liver, head kidney, and spleen were sampled from 5% fish randomly for pathogens examination as reported before “ The authors have done the spleen transcriptome, whereas have sampled also other organs. They should reported also the miRs expressions (for example by qRT-PCR) in the head kidney. The head kidney in fish is both primary (for B-lymphocytes/granulocytes) and secondary (T-lymphocytes, thrombocytes, macrophages) and it is a central lymphoid organ showing a strong activity after NNV-Virus infection [Development and Evaluation of a Bicistronic DNA Vaccine against Nervous Necrosis Virus in Pearl Gentian Grouper (Epinephelus lanceolatus × Epinephelus fuscoguttatus). Vaccines 2022, 10, 946. https://doi.org/10.3390/vaccines10060946; Bandín I, Souto S. Betanodavirus and VER Disease: A 30-year Research Review. Pathogens. 2020 Feb 9;9(2):106. doi: 10.3390/pathogens9020106. PMID: 32050492; PMCID: PMC7168202]. These articles should be read and cited by the authors because are significant for the manuscript.

Lines 182-193/277-292: “2.9. Apoptosis Assay by Fluorescence Microscope and Flow Cytometry...” The authors have done only a test without controlling the possible apoptosis arrest. Furthermore, Annexin V/PI can mark both apoptotic and necrotic cells, so is not exclusive to apoptotic cells. Moreover, Annexin is one of the first steps of apoptosis activation but is not definitive for its completion. The process could be arrested by some mechanisms, like NF-kB/IkB (nuclear factor kappa-light-chain-enhancer of activated B cells/ inhibitor kappa-light-chain-enhancer of activated B cells) or HSP70. Thus, the test used by the authors counts only the cells “activated versus apoptosis process” but is not able to count the effective apoptosis [see these papers Beere HM (2004) ‘‘The stress of dying’’: the role of heat shock proteins in the regulation of apoptosis. J Cell Sci 117:2641–2651; Veterinary Immunology and Immunopathology 162 (2014) 83–95; Front. Cell Dev. Biol., 07 August 2019, Sec. Mitochondrial Research https://doi.org/10.3389/fcell.2019.00154]. In this view, the authors need necessarily to do further analysis for HSP70 expression contemporary to their apoptotic test.

Round 2

Reviewer 1 Report

Authors made significant efforts to address all the comments.

Reviewer 2 Report

The authors have followed all the criticism and the manuscript can be published